# Effectiveness and Safety of a New Nutrient Fixed Combination Containing Pollen Extract Plus Teupolioside, in the Management of LUTS in Patients with Benign Prostatic Hypertrophy: A Pilot Study

**DOI:** 10.3390/life12070965

**Published:** 2022-06-27

**Authors:** Lucia Muraca, Antonio Scuteri, Elisabetta Burdino, Gianmarco Marcianò, Vincenzo Rania, Luca Catarisano, Alessandro Casarella, Erika Cione, Caterina Palleria, Manuela Colosimo, Antonio Cutruzzolà, Cristina Vocca, Emanuele Basile, Rita Citraro, Gabriella Marsala, Giulio Di Mizio, Giovambattista De Sarro, Luca Gallelli

**Affiliations:** 1Department of Primary Care, ASP Catanzaro, 88100 Catanzaro, Italy; lalumuraca@gmail.com (L.M.); scuteranto@gmail.com (A.S.); elisabetta.burdino@gmail.com (E.B.); 2Department of Health Sciences, University “Magna Græcia” of Catanzaro, 88100 Catanzaro, Italy; gianmarco.marciano@libero.it (G.M.); raniavincenzo1@gmail.com (V.R.); lucacatarisano@gmail.com (L.C.); al.cas1993@gmail.com (A.C.); a.cutruzzola@unicz.it (A.C.); cristina_vocca@live.it (C.V.); emanuele.basile1082@virgilio.it (E.B.); citraro@unicz.it (R.C.); desarro@unicz.it (G.D.S.); 3Department of Pharmacy, Health and Nutritional Sciences, Department of Excellence 2018-2022, University of Calabria, Ed. Polifunzionale, Arcavacata di Rende, 87036 Rende, Italy; erika.cione@unical.it; 4GalaScreen Laboratories, University of Calabria, Ed. Polifunzionale, Arcavacata di Rende, 87036 Rende, Italy; 5Operative Unit of Clinical Pharmacology, Mater Domini University Hospital, 88100 Catanzaro, Italy; palleria@unicz.it; 6Operative Unit of Microbiology and Virology, Pugliese Ciaccio Hospital, 88100 Catanzaro, Italy; manuelacolosimo@hotmail.it; 7FAS@UMG Research Center, Department of Health Science, School of Medicine, University of Catanzaro, 88100 Catanzaro, Italy; 8Dipartimento del Farmaco, U.O.C. di Farmaceutica Convenzionata, 95100 Catania, Italy; marsalagabriella@libero.it; 9Forensic Medicine, Department of Law, Economy and Sociology, University “Magna Græcia” of Catanzaro, 88100 Catanzaro, Italy; giulio.dimizio@unicz.it; 10Medifarmagen SRL, University of Catanzaro and Mater Domini University Hospital, 88100 Catanzaro, Italy

**Keywords:** benign prostatic hyperplasia, LUTS, nutrients, pollen extract, teupolioside, clinical care, drug safety

## Abstract

Benign prostatic hyperplasia (BPH) is a common cause of male lower urinary tract symptoms (LUTS) that can reduce quality of life. Even if several drugs can be used in its treatment, the development of adverse drug reactions (ADRs) represents the most common cause of low adherence. In the present study, we evaluate both the efficacy and the safety of a new nutrient fixed combination of Pollen Extract plus Teupolioside, named Xipag^®^, in patients with LUTS. We conduct a pilot single center open label clinical study between 1 March 2020 and 30 June 2020 in patients with BPH referred to general practitioner’s ambulatories. Male patients > 45 years, sexually active, with clinical symptoms of LUTS and with a diagnosis of HPB were enrolled and received one tablet/day of Xipag^®^ (T0), for three months (T1: end of treatment). The IPSS and IIEF-5 questionnaires were carried out at T0 and T1 and represent the first end point, whereas the primary safety end point was considered the absence of ADR or of drug–drug interactions related to Xipag^®^ administration. During the study period, 25 subjects aged 43 to 76 years (mean 62.7 ± 9) were enrolled and completed the study. The clinical evaluation in T1 documented that Xipag^®^ induced a statistically significant improvement (*p* < 0.01) in symptoms, as documented by the IPSS questionnaire (range 22.7–88.9; mean 55.2 ± 23.6), without the development of ADRs. In conclusion, this is the first real-world study that showed the efficacy and the safety of Xipag^®^ in the BPH patients with LUTS.

## 1. Introduction

Benign prostatic hyperplasia (BPH) affects more than 50% of men between the ages of 50 and 60 years and about 75% of those between 60 and 90 years [1]. BPH is a common cause of lower urinary tract symptoms (LUTS) in men, and it is able to reduce quality of life.

Even if several mechanisms have been suggested to be involved in the development of BPH and LUTS (e.g., altered ratio between plasma estrogens and testosterone; reduced levels of oxido-reductase with the increase in dihydrotestosterone; increased levels of oxidative stress that induce lipid peroxidation, resulting in an increase in nicotinamide adenine dinucleotide (NAD) and NAD in reduced form), multiple pieces of clinical evidence suggest that chronic inflammation seems to play a determining role not only as a pathogenetic factor but also as a predisposing factor for the progression of the disease and the severity of symptoms [2].

In fact, recently, Fibbi et al. [3], reviewing literature data, documented that the prostate contains several cells of the immune system (e.g., T and B lymphocytes and macrophages). Therefore, in the presence of a noxa, these cells release interferon type II (IFN-II), interleukin (IL)- 2, IL-6, IL-8, IL-17 6,8, and 17, and free radicals that increase the inflammation (Figure 1).


*IL, interleukin; ROS, reactive oxygen species.*


The standard of care is represented by α-adrenergic-receptor antagonists with or without the 5α-reductase inhibitor, able to induce a decrease in symptom progression and acute urinary retention [4]. Moreover, Austin et al. [5] suggested that, in some patients, the lack of the response to 5 alpha reductase inhibitors (5ARIs) could be related to the overexpression of the nuclear transcription factor Kb (NF-Kb), which is associated with an increased severity of the disease. 

The development of adverse drug reactions (ADRs) reduces the adherence to the treatment [6].

Moreover, in an experimental study, Funahashi et al. [7] suggested that the prostate’s inflammation can induce a bladder sensitization with bladder hyper-reactivity (Figure 2).

Therefore, the management of inflammation plays a role in the pharmacological approach in patients with BPH and LUTS.

In recent years, the use of nutraceutical agents increased [8,9,10] and some of these (e.g., *Cucurbita pepo*, *Hypoxis rooperi*, *Pygeum africanum*, *Serenoa repens*, *and Urtica dioica*) have been commercialized in Italy for the management of BPH [11].

In particular, a new nutrient fixed combination of pollen extract (Graminex^®^ G96; 500 mg) plus teupolioside (Teupol 25P^®^; 60 mg), named Xipag^®^ (IDI Integratori Dietetici Italiani S.r.l., Aci Bonaccorsi (CT), Italy), is being studied for BPH management. Teupolioside is a biotechnological active compound, obtained from immortalized cell cultures of *Ajuga reptans* (novel food). Teupol 25P^®^ (60 mg) contains 15 mg of phenylpropanoid teupolioside and guarantees a titration of at least 25% teupolioside. This compound has obtained registration as Novel Food.

Teupol 25P^®^, through its metabolite phenylpropanoid teupolioside, reduces the prostate production of dihydrotestosterone from testosterone, not as a direct inhibitor of 5 alpha reductase but through the oxidation of coenzyme NADPH. Preliminary in vitro data ascribe “a potency” in reducing the dihydrotestosterone production higher than *Serenoa repens* and like finasteride. This would lead to a decrease in prostatic volume over time. A pronounced anti-inflammatory activity has also been reported.

Graminex^®^ flower extract is a standardized extract, in a water-soluble (180 mg) and fat-soluble (12 mg) fraction of pollen from Secale cereale L. It has a complex and complete chemical composition with selective action on the prostate district, where it has a marked anti-inflammatory, anti-edema, and anti-radical action [ORAC value (288 μmol TE/g)].

Contains 21 amino acids, enzymes, coenzymes, minerals, carotenoids, flavonoids, phenols (including carvacrol, rutin, chlorogenic acid), fatty acids (including alpha linoleic acid), trace elements, and water- and fat-soluble vitamins.

The clinical efficacy of pollen extract in the treatment of prostatitis, also associated with pelvic pain [12], has been widely demonstrated in controlled clinical trials [13,14]. In an experimental study, Nagashima et al. [15] investigated the effects of pollen extract on inflammatory cytokines (IL-1 beta, IL-6 and TNF-alpha) in an animal model of prostatic disease treated with pollen extract (630 and 1260 mg/kg, p.o.) or testosterone (2.5 mg/kg, s.c.). In this study, the authors documented that pollen extract induced a dose-dependent decrease in cytokines with a normalization of both glandular inflammation and stromal proliferation. Wagenlehner et al. [16] reported that the pollen extract acts on the smooth muscles of the uro-genital area.

More recently in a randomized clinical trial, Matsukawa et al. [17] evaluated the effect of a 3-month treatment with pollen extract (n. 42 men) or tadalafil (n. 45 men) in men with chronic prostatitis/chronic pelvic pain syndrome (CP/CPPS) and lower urinary tract symptoms (LUTS). In this study, the authors documented that both pollen extract and tadalafil significantly improved chronic pelvic pain, even if the pollen extract was more effective than tadalafil in the improvement of pelvic pain and discomfort.

Finally, Locatelli et al. [18] in a chromatography study documented a significant presence of phenolic patterns (rutin, chlorogenic acid, and carvacrol) in Graminex^®^ G96, which justify a multi-target action on the biochemical events that trigger the inflammatory process in the prostate. In fact, rutin, chlorogenic acid, and especially carvacrol have an inhibitory effect on the nuclear factor kappa-light-chain-enhancer of activated B cells (NF-κB). The authors, confirmed in an in vitro study in prostatic cells culture, that the pollen extract significantly inhibits the pathway of NF-κB with a consequent decrease in the cellular synthesis of pro-inflammatory substances and free radicals. Therefore, this new combination could represent a novelty in the treatment of LUT in patients with BPH because it contains two compounds able to block the pathways involved in the development of clinical manifestations in these patients (see Figure 3 for the possible mechanisms of action).

Pollen exerts its effects reducing the production of inflammatory cytokines. NF-kB inhibition further reduces inflammation, resulting in the decrease in this process and of stromal proliferation in the prostate. An action on pelvic pain was also documented. Teupolioside has been found to turn off the inflammatory process and to inhibit DHT, the testosterone derivate responsible of the pathologic mechanism.

DHT, dihydrotestosterone; NADPH, nicotinamide adenine dinucleotide phosphate; NF-kB, nuclear factor kappa-light-chain-enhancer of activated B cells.

The aim of the present study is to evaluate the efficacy and the safety of this new fixed combination of pollen extract (Graminex^®^ G96; 500 mg) plus teupolioside (Teupol 25P^®^; 60 mg) in patients with LUTS.

## 2. Materials and Methods

### 2.1. Study Design

We conducted a pilot single center open label clinical study between 1 March 2020 and 30 June 2020 in patients with BPH referred to a general practitioner’s ambulatory. The study, approved by the Ethics Committee of Calabria Centro (number 22/2019), was carried out according to the Good Clinical Practice guidelines and with the ethical principles of the Declaration of Helsinki. Before the beginning of the study, all participants signed a written informed consent.

### 2.2. Experimental Protocol

Patients with a clinical and instrumental diagnosis of moderated LUTS related to BPH were recruited. At the time of enrollment (T0), the physician administered the International Prostatic Symptom Score (IPSS) and International Index of Erectile Function (IIEF-5) questionnaire. A blood sample was collected to evaluate plasma concentrations of prostate specific antigen (PSA). All patients who met the inclusion criteria were enrolled and received 1 tablet/day of Xipag^®^ (IDI Integratori Dietetici Italiani S.r.l., Aci Bonaccorsi (Catania), Italy) (T0), for 3 months (T1: end of treatment), when the patients completed the IPSS and IIEF-5 questionnaires. The IIEF-5 questionnaire was used to rule out patients with erectile dysfunction and to assess the effect of therapy on sexual function at the end of the study.

### 2.3. Inclusion and Exclusion Criteria

In this study, male patients > 45 years, sexually active, with clinical symptoms of LUTS and a diagnosis of HPB were included. Patients with allergies to one or more components of XIPAG^®^ were excluded.

### 2.4. End Points

The primary clinical end point was the statistically significant improvement (*p* < 0.05) of quality of life at T1 compared to T0, in terms of changes in IPSS.

The secondary clinical end point was the statistically significant change (*p* < 0.05) in IIEF-5 score and/or PSA value at T1 vs. T0.

Finally, the primary safety end point was considered the absence of ADRs or drug-drug interactions (DDIs) related to Xipag^®^ administration.

Consistent with our previous article, we used the Naranjo scale and the Drug Interaction Probability Scale to evaluate the development of ADRs or DDIs, respectively [19,20,21,22].

The persistence of symptoms in T1, and/or the development of ADRs, that could lead to the discontinuation of treatment, were defined as clinical failure.

### 2.5. Drug Adherence

At the end of the study, adherence to the treatment was calculated using a formula based on the number of tablets consigned at the time of admission and the number of tablets returned unused at the end of the study.

### 2.6. Drugs

Teupol 25P is a biotechnological extract obtained by Ajuga reptans and guarantees a titration of at least 25% teupolioside and has obtained registration as Novel Food. Teupol 25P (60 mg) contains 15 mg of phenylpropanoid.

### 2.7. Statistical Analysis

At baseline, the independent sample 2-tailed *t*-test was used to compare variables. For categorical parameters, chi-square test was applied. Changes from baseline to end of therapy were analyzed using ranked one-way analysis of variance (ANOVA) with a term for treatment group. Both Kruskal–Wallis and sign tests were used for non parametric evaluation (IPSS and IIEF-5 scores). All data are expressed as mean ± standard deviation (SD). The threshold of statistical significance was set at *p* < 0.05. All reported *p*-values are two-sided. All statistical analyses were performed by using SPSS 21.0 (IBM Corporation, Armonk, NY, USA).

## 3. Results

### 3.1. Patients

During the study period, 25 subjects aged 43–76 years (average 62.7 ± 9 years) were enrolled and completed the study. At the time of recruitment, in all patients, the presence of BPH was confirmed using the IPSS questionnaire (range 5–26; mean 13.3 ± 6.1). In 22 patients (88%), PSA values were in the normal range (Table 1). Values greater than 2.5 ng/mL were detected in two patients (8%) aged 64 and 76.

### 3.2. Effect of the Treatment on Clinical Parameters

The clinical assessment performed at the end of the study (T1) showed that the treatment with Xipag^®^ induced a statistically significant improvement (*p* < 0.01) in symptoms, as demonstrated by the IPSS questionnaire (decrease T3 vs. T0: range 22.7–88.9%; mean 55.2 ± 23.6%) (*χ*^2^ = 26.338 > *χc*^2^ = 3.841) (Figure 1). In only two patients (8.7%), aged 49 and 66 years, Xipag^®^ did not induce any change in the IPSS scale (patients n. 21 and 23 in Figure 4). However, these two patients showed a statistically significant decrease (*p* < 0.01) in the frequency of nicturia (from 5 to 2 episodes) (primary end point). In contrast, the treatment with Xipag^®^ (T1) did not modify the PSA values (secondary end point) and did not improve the erectile function (secondary outcome), as demonstrated by the IIEF-5 questionnaire (T0: mean 16.2 ± 6.8; T1: mean 16.2 ± 7) (Figure 5).

When we stratified the patients by age, before and after 65 years, we documented that the age induces a significant increase (*p* < 0.01) in IPSS and a significant decrease (*p* < 0.01) in IIEF-5 at both T0 (*p* = 0.003) and T1 (*p* = 0.003) (Table 2). When we considered the effect of Xipag^®^, we documented a significant decrease in IPSS values without difference between the groups 45–65 and 66–75 years (Figure 6) (T0: *p* = 0.52; T1: *p* = 0.34). In contrast, in IEFF-5, when we considered the effects of a 3-month treatment with Xipag^®^, we did not record a significant age-related difference (Figure 7).

### 3.3. ADRs Related to XIPAG

During the study period, we did not record ADRs or DDIs (primary safety end point), whereas we recorded complete adherence (100%) to the treatment and full compliance with the experimental protocol.

## 4. Discussion

In the present study, we reported for the first time the efficacy and the safety of a new nutraceutical (Xipag^®^) containing Teupolioside (Teupol^®^ 60 mg) and Pollen Extract (Graminex^®^ G96 500 mg) used for the treatment of LUTS in patients with HPB.

Teupolioside is a phenylpropanoid glycoside that inhibits the release of reactive oxygen species from the whole blood leukocytes [23]. Furthermore, at prostatic level, Teupolioside inhibits the transformation of testosterone into dihydrotestosterone, the release of pro-inflammatory cytokines (i.e., IL-1 beta and TNFalpha), and the expression of both total Cyclooxygenase (COX) and COX-2.

Graminex^®^ G96 is a pollen extract belonging to the Poaceae family. The bioactive component consists of a pool of amino acids, enzymes, coenzymes, minerals, carotenoids, flavonoids, fatty acids (including alpha linoleic acid), and above all, phenolic patterns. The latter are attributed a particular anti-inflammatory and apoptotic activity at the prostatic level, through the modulation (inhibitory activity) of the biochemical pathway, which results in the activation of the NFK light chain enhancer of activated B cells.

In the present study, we documented that in HPB patients with LUTS, a 3 month treatment with Xipag^®^ induced a significant improvement in clinical symptoms, as documented by the IPSS score, without any difference for age. These effects are related to the mechanism of action of each component of the nutrients. In a recent review, Cicero et al. [24] documented that several compounds reduce the secretion of cytokines. In agreement, a clinical study documented that a mixture containing lycopene induced a significant decrease in IPSS score, pollakiuria, and nocturia [25].

In addition, pollen extract inhibits the secretion of several cytokines with a similar effect of diclofenac and indomethacin and with an activity 10 fold higher than that of aspirin [26]; therefore, it has been indicated in the guidelines of the European Association of Urology for the management of chronic pelvic pain in patients with prostatitis (VOCE 8).

Finally, it has been demonstrated that beta-sitosterol inhibits testosterone conversion by inhibiting aromatase and 5α-reductase [27], thereby improving urinary symptoms [28].

Therefore, we can postulate that the effects of Xipag^®^ on LUTS are related to the synergic association of the two phytocompounds that occurs with the multitarget activity on the prostate area. These effects successfully counteract irritative/inflammatory and obstructive symptoms in BPH patients.

Therefore, this new compound could represent a novelty in the treatment of LUTS in HBP patients because the single compound is able to block the pathway of cytokines and chemokines involved in this clinical manifestation.

Moreover, these mechanisms of action explain the absence of the effects on IEFF; in fact, although erectile dysfunction is common in patients over 65 years of age, it may be related to vascular diseases rather than inflammatory or hormonal diseases. Moreover, the tested nutrient does not have vasodilator effects, and this could explain the lack of efficacy in IEFF. Finally, an important point is that the observed effects are not age related. Moreover, this is the first real-world study that documented that the treatment with this new nutrient for 3 months did not induce any ADR or DDI, suggesting the tolerability of the supplement. Its effectiveness is underlined, at the end of the treatment, by the demand for urgent care of enrolled patients who have benefited significantly from Xipag^®^ treatment.

It is important to underline that, in agreement with our recent article [24], the safety of nutrients represent a major concern also related with the informed consent. In the presence of a treatment with nutrients, adequate information must be provided to the patient regarding the scientific evidence that prompts the practitioner to prescribe a nutraceutical, any side effects, and the real expectations of treating the pathology through the administration of nutraceuticals. Consistent with this, we informed the patients about the effects of each component of the compound, and we obtained the signed consent.

## 5. Conclusions

The management of LUTS in BPH patients represents a clinical relevance for urologist and family practitioners. Even if several compounds are in the market with the indication for BPH, some of these are not able to reduce the clinical symptoms of LUTS. Therefore, a combination of nutrients able to block the inflammatory pathway with a multitarget activity could represent a novelty for the treatment of LUTS. In agreement with this, in this study we documented for the first time the clinical effect of a new nutraceutical (Xipag^®^), suggesting that it is effective and safe in the management of LUTS in patients with BPH. However, this study has a limitation represented by the low number of patients enrolled; therefore, we hope that other clinical trials in a large number of patients can be performed in order to validate these results.

## Figures and Tables

**Figure 1 life-12-00965-f001:**
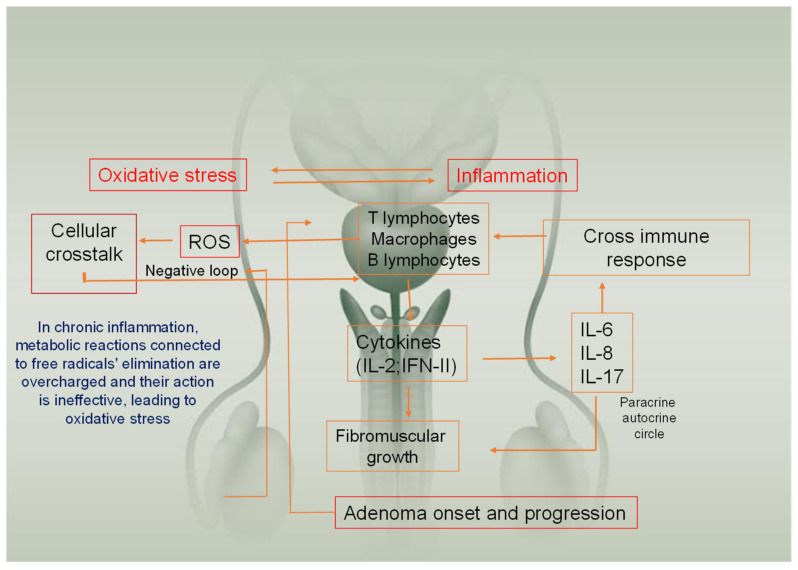
Mechanisms involved in BPH: the relation between oxidative stress and inflammation play a role in both pathogenesis and progression of Benign Prostatic Hypertrophy. Chronic inflammation (through cytokines release) increases fibromuscular growth. This process continuously produces reactive oxygen species, overcharging enzymes deputed to their removal, and then enhancing the damage.

**Figure 2 life-12-00965-f002:**
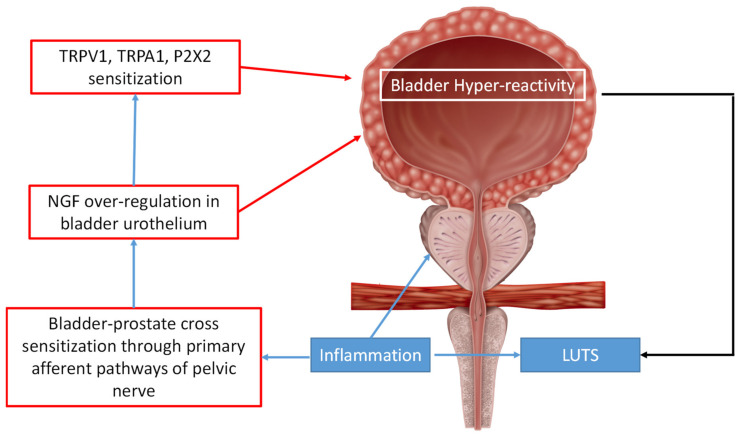
Role of prostate’s inflammation in bladder hyperreactivity. Prostate inflammation determines obstructive symptoms in urinary tract. This clinical condition has harmful consequences in bladder. Sensitization of bladder-prostate through pelvic nerve, increases bladder activity (through detrusor overstimulation), propelling the militainment of a chronic condition. Similar effects may be related to the overregulation of NGF (related to pelvic nerve overactivation) and channel receptors TRPV1, TRPA1, and P2 × 2 (connected to neurons hyperexcitability). *LUTS, lower urinary tract symptoms; NGF, nerve growth factor; TRPA, transient receptor potential ankyrin; TRPV, transient receptor potential vanilloid; P2X2: purinoceptor 2*.

**Figure 3 life-12-00965-f003:**
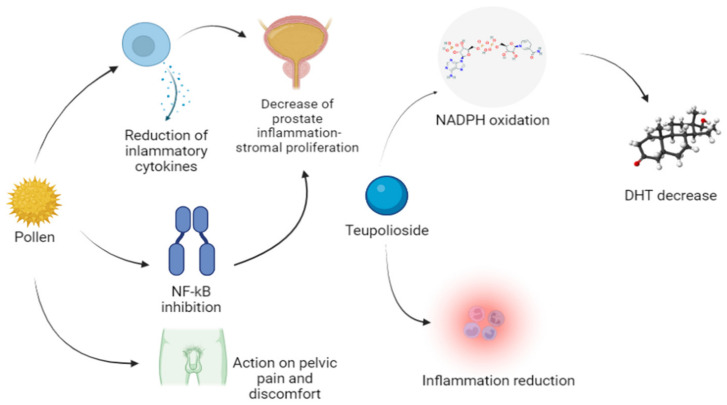
Xipag’s pharmacodynamic.

**Figure 4 life-12-00965-f004:**
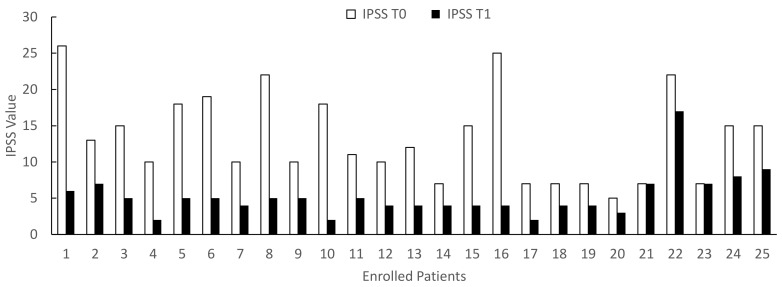
IPSS (International Prostatic Symptom Score) values in enrolled patients evaluated before (T0) and after 3 months of treatment with Xipag^®^ (T1). Data represent the absolute values recorded in all enrolled patients (n. 25). For each patient *p* < 0.01. Only in two patients (n. 21 and 23), we failed to report a significant improvement in IPSS values.

**Figure 5 life-12-00965-f005:**
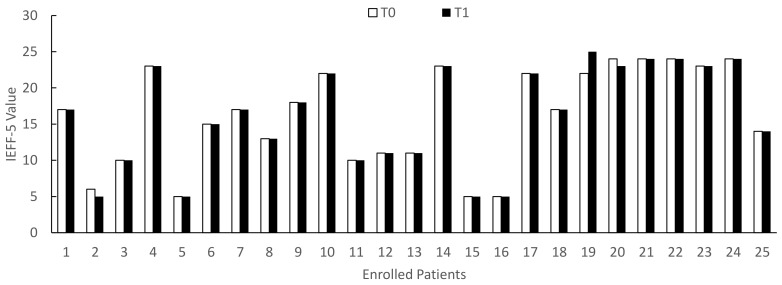
IEFF-5 (International Index of Erectile Function questionnaire) values in enrolled patients evaluated before (T0) and after 3 months of treatment with Xipag^®^ (T1). Data represent the absolute values recorded in all enrolled patients (n. 25).

**Figure 6 life-12-00965-f006:**
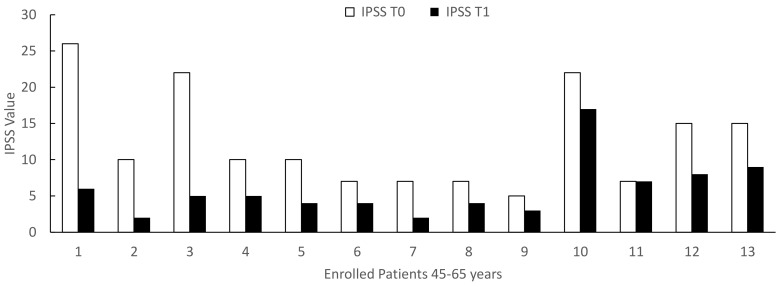
IPSS (International Prostatic Symptom Score) values in enrolled patients 45–65 upper and 66–76 down, evaluated before (T0) and after 3 months of treatment with Xipag^®^ (T1). Data represent the absolute values recorded in all enrolled patients (n. 13 and 12, respectively). For each patient *p* < 0.01. Only in two patients (n. 11 in Group 45–65 years and n 12 in Group 66–76 years), we failed to report a significant improvement in IPSS values.

**Figure 7 life-12-00965-f007:**
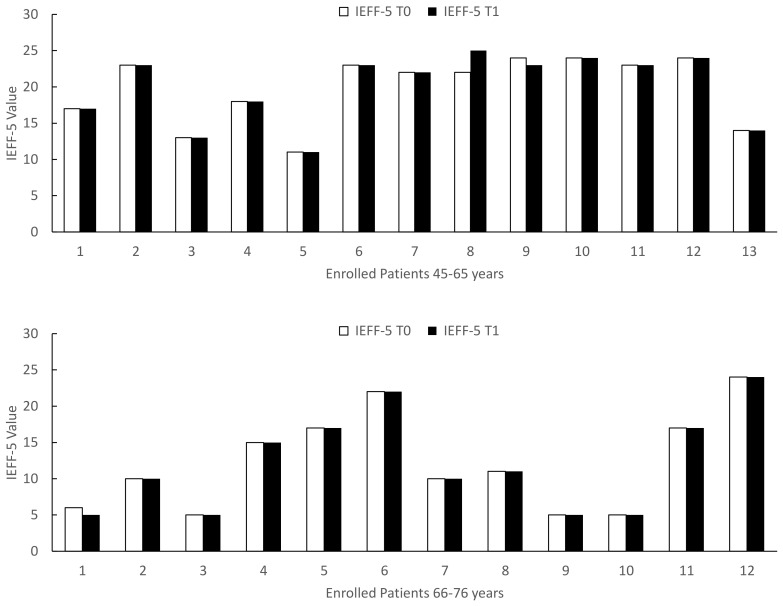
IEFF-5 (International Index of Erectile Function questionnaire) values in enrolled patients 45–65 upper and 66–76 down, evaluated before (T0) and after 3 months of treatment with Xipag^®^ (T1). Data represent the absolute values recorded in all enrolled patients (n. 13 and 12, respectively). For each patient *p* < 0.01.

**Table 1 life-12-00965-t001:** Demographic characteristics of enrolled patients at the time of admission (T0). Data are expressed as mean ± standard deviation. COPD: chronic obstructive pulmonary diseases. IPSS: International Prostatic Symptom Score; IIEF-5: International Index of Erectile Function questionnaire. PSA: prostate specific antigen.

Demographics	Values
Range (years)	43–76
Number of enrolled patients	25
Age (years)	62 ± 9.3
IPSS (T0)	13.3 ± 6.1
PSA	1.4 ± 1
IIEF-5 (T0)	16.2 ± 6.8
COPD/Asthma	3/1
Blood Hypertension	11
Diabetes	4

**Table 2 life-12-00965-t002:** Effects of Xipag^®^ in enrolled patients (n. 25) with LUTS stratified for age. Data are expressed as mean ± standard deviation (SD). COPD: chronic obstructive pulmonary diseases. IPSS: International Prostatic Symptom Score; IIEF-5: International Index of Erectile Function questionnaire. PSA: prostate specific antigen. * *p* < 0.01.

Parameters	45–65 Years	66–76 Years	*p*
Number of patients	13	12	-
Age (years)	62 ± 9.3	69.8 ± 4	*
IPSS (T0)	12.5 ± 6.9	14.2 ± 5.3	*
PSA	1.4 ± 1	1.5 ± 1.2	-
IIEF-5 (T0)	19.8 ± 4.7	12.3 ± 6.7	*
COPD/Asthma	2	3	-
Blood Hypertension	3	8	*
Diabetes	2	2	-

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
