# Peer review of "Effectiveness and Safety of a New Nutrient Fixed Combination Containing Pollen Extract Plus Teupolioside, in the Management of LUTS in Patients with Benign Prostatic Hypertrophy: A Pilot Study"

_life, 2022, doi:10.3390/life12070965_

Round 1
Reviewer 1 Report
Dear Authors,
The present study evaluates the efficacy of a nutrient combination of Pollen Extract plus Teupolioside, named Xipag®, in patients with LUTS. The research subject is interesting and brings scientific important data in the field, as it deals with a subject that is currently of great interest. Some changes of the manuscript should nevertheless be performed in order to improve its quality. Following specific changes should thus be performed:
Minor changes
I think that the combination of terms “flower pollen extract” is too much…is there another type of pollen, except the one of flowers?
Please add paternity of all types of species, the first time they appear in text. Also, please correct names of species: fisrt letter should be capital, second not. All names of species must be italic.
Major changes
Introduction: This section needs serious changes. The whole part needs a proper introduction into context and explanations on the state of the art on the subject. Some phrases are unclear, please rephrase. It is not clear what the described nutrient combination refers to, what does it contain, how the individual components it contains were obtained. Also, this part should contain information regarding similar existing studies in literature and, in comparison, authors should emphasize the novelty and originality of their study. Please explain and cite each of these studies. Background of the study is also not clear. The purpose of the study needs to be rephrased to become clearer. Please add further information and justifications and modify accordingly. It is absolutely clear that the study is not well documented as too little references are cited. Once again, this section needs serious changes, it is too short and does not offer a proper introduction into the large context. Please offer a rationale for choosing this nutraceutical combination and offer more details about it in the context of your study.
Discussions: Here you should emphasize novelty and originality of the present study once again. You need to compare your results with the ones obtained by other authors and you need to highlight what you bring in novelty compared to these. Clearly you need to offer more details and develop this part in order to bring consistency to your study.
Conclusions: This section should be added. Please offer perspectives of your study, it is not ok to conclude a study in Discussions.
All these suggested changes should be performed in order to bring further improvements to the manuscript.
Author Response
Dear Reviewer 1
Thank you for your comments that we considered for our revised version
Minor changes
I think that the combination of terms “flower pollen extract” is too much…is there another type of pollen, except the one of flowers?
Answer: in agreement with your suggestion, we changed it
Please add paternity of all types of species, the first time they appear in text. Also, please correct names of species: fisrt letter should be capital, second not. All names of species must be italic.
Answer: in agreement with your suggestion, we revised this point
Major changes
Introduction: This section needs serious changes. The whole part needs a proper introduction into context and explanations on the state of the art on the subject. Some phrases are unclear, please rephrase. It is not clear what the described nutrient combination refers to, what does it contain, how the individual components it contains were obtained. Also, this part should contain information regarding similar existing studies in literature and, in comparison, authors should emphasize the novelty and originality of their study. Please explain and cite each of these studies. Background of the study is also not clear. The purpose of the study needs to be rephrased to become clearer. Please add further information and justifications and modify accordingly. It is absolutely clear that the study is not well documented as too little references are cited. Once again, this section needs serious changes, it is too short and does not offer a proper introduction into the large context. Please offer a rationale for choosing this nutraceutical combination and offer more details about it in the context of your study.
Answer: in agreement with your suggestion, introduction was rewritten
Discussions: Here you should emphasize novelty and originality of the present study once again. You need to compare your results with the ones obtained by other authors and you need to highlight what you bring in novelty compared to these. Clearly you need to offer more details and develop this part in order to bring consistency to your study.
Answer: in agreement with your suggestion, we revised the discussion
Conclusions: This section should be added. Please offer perspectives of your study, it is not ok to conclude a study in Discussions.
All these suggested changes should be performed in order to bring further improvements to the manuscript.
Answer: in agreement with your suggestion, we added this section
Reviewer 2 Report
The manuscript submitted by Muraca et, al. is a study which tried to evaluate the Effectiveness and safety of Xipag, in the management of LUTS in patients with BPH, the manuscript is well written but the findings are very preliminary, I believe that this study have major concerns:
1. On what basis did the investigator decide the duration of the study (3 months)?
2. Why did they chose the dosage used (1 tablet/day of Xipag) ? specially that Graminex® G96; 500 mg can be used up to three times per day.
3. Why there is no negative control group to compare with?
4. Why there is no groups that took only flower pollen extract (Graminex® G96; 500 mg) or teupolioside to compare with?
5. In figure 1 the IPSS value was 26 maximum while in the text the range is between 22.7-88.9
6. In page 5 first paragraph the Authors stated “In contrast, when we considered the effects of a 3-month treatment with Xipag ® we did not record a significant difference age-related (Figure 4).”, the authors should clearly mention that they talk aout IEFF-5
Author Response
Dear Reviewer 2
We revised our manuscript in agreement with your comments and we send you the final version with our point by point answers
- On what basis did the investigator decide the duration of the study (3 months)?
Answer: The time of the study was evaluated considering the mean time of improvement of symptoms with traditional drugs and also in agreement with other manuscript as reported in references
- Why did they chose the dosage used (1 tablet/day of Xipag) ? specially that Graminex® G96; 500 mg can be used up to three times per day.
Answer: We used the dosage in agreement with manufacturers data. Xipag is a fixed combination of Graminex plus Teupolioside. Theferore, using the synergic pharmacological effect we reduced the dosage and the time of administration in order to obtain ad high safety and a good adherence.
- Why there is no negative control group to compare with?
Answer: in Italy it is not possible to have an untreated group.
- Why there is no groups that took only flower pollen extract (Graminex® G96; 500 mg) or teupolioside to compare with?
Answer: data regarding the single component have been published by several groups, and now we evaluated only the synergistic activity of this new compound.
- In figure 1 the IPSS value was 26 maximum while in the text the range is between 22.7-88.9.
Answer: It is correct, we revised this point. The range 22.7-88.9 represents the difference in % between T1 and T0.
- In page 5 first paragraph the Authors stated “In contrast, when we considered the effects of a 3-month treatment with Xipag ® we did not record a significant difference age-related (Figure 4).”, the authors should clearly mention that they talk aout IEFF-5
Answer: This point has been added
Reviewer 3 Report
The topic of this clinical study is interesting. The reviewer feels it can be accepted after major amendments.
1) The source of Xipag should be indicated clearly.
2) This clinical study lacks placebo / blindness. This is the major limitation.
3) The statistical method is invalid. For score data, the authors have to use nonparametric approach. t-test requires continuous data.
4) How can the authors assure there is absence of DDI? Without pharmacokinetic study, no one can make such claim. Any literature reporting Teupolioside cause DDI?
Author Response
Dear Reviewer 3
We revised our manuscript in agreement with your comments and we send you the final version with our point by point answers
- The source of Xipag should be indicated clearly.
Answer: we added this information
2) This clinical study lacks placebo / blindness. This is the major limitation.
Answer: It is correct but in Italy it is not possible to have a placebo treated group. In order to improve the safety we did not perform a blindness study.
3) The statistical method is invalid. For score data, the authors have to use nonparametric approach. t-test requires continuous data.
Answer: we reevaluated the data considering the nonparametric approach using Kruskal-Wallis Test and Sign Test
4) How can the authors assure there is absence of DDI? Without pharmacokinetic study, no one can make such claim. Any literature reporting Teupolioside cause DDI?
Answer: It is correct we failed to have a adverse drug reaction related to xipag or to other drugs usually treated. Therefore, commonly we used pharmacokinetic study and therapeutic drug monitoring (TDM) in patients that during a common treatment develop an adverse drug reaction. In these patients, in order to evaluate the presence of a drug interaction, we perform the TDM. In the present study we did not record the development of ADRs related to Xipag or related to DDI.
Round 2
Reviewer 1 Report
Dear Authors,
The present study evaluates the efficacy of a nutrient combination of Pollen Extract plus Teupolioside, named Xipag®, in patients with LUTS. The authors performed most of the suggested changes after the first round of review. Following specific changes should still be performed:
Minor changes
Paternity is not added for all types of species the first time they appear in text.
Major changes
Introduction: It is still not clear what the described nutrient combination refers to, what does it contain, how the individual components it contains were obtained. Also, authors should still emphasize the novelty and originality of their study, by comparison with similar studies that are cited. The purpose of the study still needs to be rephrased as it is not clear. At the same time, some figures were added in this part: are they novel and original (performed by authors) or are they taken from different references? If so, please cite in their titles.
Discussions: Novelty and originality of the present study is still not emphasized. You need to highlight what you bring in novelty compared to similar studies that are cited.
Conclusions: Please develop this section, 3 lines are not enough to conclude your study. Please add perspectives.
All these suggested changes should be performed in order to bring further improvements to the manuscript.
Author Response
Dear Referee
We revised again the manuscript in agreement with your suggestions in order to clarify it
Minor changes
Paternity is not added for all types of species the first time they appear in text.
Answer: the paternity has been added
Major changes
Introduction: It is still not clear what the described nutrient combination refers to, what does it contain, how the individual components it contains were obtained.
Answer: as reported in the text, the new formulation Xipag® (IDI Integratori Dietetici Italiani S.r.l., Aci Bonaccorsi (CT), Italy) contains: teupolioside 60 mg + pollen extract 500 mg
Also, authors should still emphasize the novelty and originality of their study, by comparison with similar studies that are cited. The purpose of the study still needs to be rephrased as it is not clear.
Answer: the novelty has been added and the purpose has been clarified (see page 4, bottom)
At the same time, some figures were added in this part: are they novel and original (performed by authors) or are they taken from different references? If so, please cite in their titles.
Answer: the figures have been added because in agreement with referee’s request. These figures are original (we have done it) and have not been taken from other studies.
Discussions: Novelty and originality of the present study is still not emphasized. You need to highlight what you bring in novelty compared to similar studies that are cited.
Answer: this point has been revised in agreement with your suggestions
Conclusions: Please develop this section, 3 lines are not enough to conclude your study. Please add perspectives.
Answer: the conclusions have been revised
Reviewer 2 Report
I would like to thank the authors for their responses
Author Response
Thank you for your time and for your comments that help us to improve this manuscript
Reviewer 3 Report
The manuscript has been improved and appears to be acceptable.
Author Response
Thank you